# Entry Point Variation in the Osseous Fixation Pathway of the Anterior Column of the Pelvis—A Three-Dimensional Analysis

**DOI:** 10.3390/jpm12101748

**Published:** 2022-10-21

**Authors:** Lotte Dupuis, Laura A. van Ginkel, Luc M. Verhamme, Thomas J. J. Maal, Erik Hermans, Vincent M. A. Stirler

**Affiliations:** 1Department of Trauma Surgery, Radboud University Medical Center, 6525 GA Nijmegen, The Netherlands; 2Radboudumc 3D Lab, Radboud University Medical Center, 6525 GA Nijmegen, The Netherlands

**Keywords:** acetabular fracture, pelvic ring injury, screw insertion, entry point, anterior column, osseous fixation pathway, corridor, 3D planning, 3D imaging, computer-aided surgery

## Abstract

Fractures of the superior pubic ramus can be treated with screw insertion into the osseous fixation pathway (OFP) of the anterior column (AC). The entry point determines whether the screw exits the OFP prematurely. This can be harmful when it enters the hip joint or damages soft tissues inside the lesser pelvis. The exact entry point varies between patients and can be difficult to ascertain on fluoroscopy during surgery. The aim of this study was to determine variation in the location of the entry point. A retrospective single center study was performed at a level 1 trauma center in the Netherlands. Nineteen adult patients were included with an undisplaced fracture of the superior pubic ramus on computer tomography (CT)-scan. Virtual three-dimensional (3D) models of the pelvises were created. Multiple screws were placed per AC and the models were superimposed. A total of 157 screws were placed, of which 109 did not exit the OFP prematurely. A universally reproducible entry point could not be identified. A typical crescent shaped region of entry points did exist and was located more laterally in females when compared to males. Three-dimensional virtual surgery planning can be helpful to identify the ideal entry points in each case.

## 1. Introduction

Pelvic and acetabular fractures are severe musculoskeletal injuries, which are mostly caused by high-energy traumas. Minimal invasive fixation techniques can avoid the morbidity of open surgery in these injuries [1]. These techniques are especially relevant when considering patients with obesity as well as polymorbidity in the elderly and multiple injured patients [2,3].

Opportunities for minimal invasive fixation of the pelvis and acetabulum are conceptualized as osseous fixation pathways (OFPs). OFPs are geometrically complex corticated bony cylinders of different dimensions and orientations that can accommodate intraosseous implants [1]. The anterior column (AC) of the pelvis has an OFP that extends from the pubic symphysis to the supraacetabular lateral iliac cortex and includes the entire superior pubic ramus, anterior acetabular wall, and supraacetabular area [1]. A screw can be placed in antegrade or retrograde directions. The location of the entry point and the angulation of the screw determine whether its placement is inside the OFP or exits it prematurely. The latter can be harmful when it involves the joint or the soft tissues inside the lesser pelvis.

The location of the entry point varies between patients. Typically, intraoperative fluoroscopy is required to determine its exact location [4]. This can be a time-consuming process that involves multiplanar radiographs, and multiple attempts may be required to successfully cannulate the OFP [1]. Understanding the patient-specific anatomy of the AC could expediate screw placement into the OFP.

Only two studies have investigated the localisation of entry points into the OFP of the AC. Their approach involved spatial measurements that were reported as average positions on the pelvis. Such results are less practical during minimally invasive techniques where the soft tissue envelope remains intact. In addition, the theoretical average position of an entry point will often vary between actual patients.

We hypothesise that there is an entry zone, a collection of multiple entry points, from which screws can safely be implanted into the AC of adult patients. The aim of this study was to investigate the extent of variation in entry points. To this end we performed a three-dimensional (3D) analysis of the anatomy of AC of the pelvis and safe screw positions in the OFP.

## 2. Materials and Methods

Ethical review and approval were waived for this study by the local medical research ethics committee, due to the fact that all included computer tomography (CT)-scans were acquired during standard treatment, anonymisation and retrospective nature of the study. The requirement for patient consent was waived as all image data were anonymized and de-identified prior to analysis. This study followed a guideline named the strengthening the reporting of observational studies in epidemiology statement (STROBE) [5].

### 2.1. Data Collection

This is a retrospective single-center study performed at the Radboud University Medical Center (RUMC), a level 1 trauma center in Nijmegen, the Netherlands. Trauma patients were considered eligible for inclusion if they were treated at RUMC between 2017 and 2021. They were included when 16 years or older and sustained a fracture in the AC of the pelvis. Exclusion criteria were preoperative CT scans of the pelvis with a slice thickness of more than 1.0 mm and fracture dislocation where percutaneous screw placement would technically not be possible. All image data were pseudonymized and de-identified prior to analysis. The keys to patient-specific data were kept separate from the study data and were only accessible to the principal investigators. Patient-specific characteristics as sex, age and length were derived from the electronic medical records and collected in an Excel database (Microsoft, Redmond, WA, USA).

The primary outcome was to determine the extent of variation in entry points for screw insertion into the OFP of the AC of the pelvis. Secondary outcome measures were the diameter and length of the screws. Differences between men and women were also studied, with both anatomical variations and variations in entry point.

### 2.2. Model Reconstruction

Digital Imaging and Communications in Medicine (DICOM) files corresponding to the selected CT scans were retrieved from the Picture Archiving and Communication System (PACS). The DICOM files were used to create virtual models of the pelvis using Mimics (version 23.0, Materialise, Leuven, Belgium). The threshold values were set to identify cortical bonee, i.e., from 226 to 1504 Hounsfield units (HU). The femora and the lumbar vertebra upward of L5 were removed from the models. The models were subsequently smoothened (Smooth Mask function). The models were exported as Stereolithography (STL) files.

### 2.3. Virtual Screw Placement

The STL-files were imported into 3DMedX^®^ (version 1.2.17.0, 3D Lab Radboudumc, Nijmegen, The Netherlands). 3DMedX^®^ was used to perform the virtual screw placement in the AC. The transparency of the pelvic models was downgraded to make the intra-osseous course of the screw visible. Inlet and obturator-outlet view were used to assess screw placement inside or outside the OFP. The models of the screws had a diameter of 3.5 mm or 6.5 mm, corresponding to the diameter of commonly used screws. As many well-positioned screws as possible were placed per case. All digitally placed screws were supervised by senior authors with extensive experience in treating pelvic fractures (E. Hermans and V.M.A. Stirler). Files were uploaded in 3DMedx. The CT slices were superimposed on the 3D model. Random reslicing was carried out by scrolling through the central axis of the screw, to verify whether the screws were positioned correctly (Figure 1).

### 2.4. Analysis of Entry Zones

The variation in entry points can be compared between cases by superimposing the 3D segmentations of the different pelvic models. The models were mirrored using Meshmixer^®^ (version 3.5, Autodesk Inc, San Rafael, CA, USA) in order to obtain only right-sided segmentations. The clinically most optimal screw position was retained per case. Three different methods were used for superimposing in 3DmedX^®^. In the first method, the Iterative Closest Point (ICP) algorithm was used to match segmentations based on the anatomy. The anterior column was selected as the focus for ICP matching. In the second method, a Procrustes analysis was used for matching. Four landmarks were used for the Procrustes analysis: the entry- and exit point of the screw, the pubic tubercle and a point at the pubic crest closest to the pubic symphysis. The final method consisted of superimposing the models manually.

Male and female cases were analyzed separately, and the entry points were compared with each other. Additionally, the distance between the upper limit of the entry points and the pubic tubercle was measured using a digital ruler in 3DMedX^®^. The distance from the lower limit of the entry points to the border of the inferior ramus was measured as well. The circumference of the entry zone was measured using Circumference Calculation in 3DMedX^®^. The region of entry points was selected, and the size of the selected area was assessed using the function called Area Calculation Selection.

## 3. Results

A total of nineteen patients were included in this study. Patient characteristics were presented in Table 1.

A total of 157 virtual screws were placed. A total of 109 (69%) screws did not exit the anterior column prematurely and had entry points that were accessible through a retrograde approach. A total of 13 (8%) screws were placed in the OFP of the AC but their entry points were deemed surgically dangerous. Another 35 (22%) screws did exit the OFP prematurely (Figure 2).

Three different methods were used to determine the variation in the entry points. Superimposing the pelvic models resulted in inadequate matching when based on anatomy or on landmarks on the OFP. Matching of the pubic bones seemed adequate, but the OFPs diverted (Figure 3).

The best results were gained after manually matching the models, especially when males and females were analysed separately (Figure 4).

Subsequently, the entry points’ zone wase determined using the superimposed 3D models of males and females. A crescent-shaped zone of entry points was identified on the pubic bone. The entry zone was located more laterally in females when compared to males (Figure 5). A universally reproducible entry point could not be identified. The same findings were true for antegrade screw placement.

A quantitative analysis of the entry zones was performed (Figure 6 and Table 2). In most cases, the AC accommodated a 6.5 mm screw. Only in 1 out of 10 males was a 3.5 mm screw placed. A 3.5 mm screw was placed in 3 out of 9 females. The median screw length was 122.0 mm (IQR 13.0 mm).

## 4. Discussion

Knowledge of appropriate entry points can be helpful in the expedient and safe placement of screws into OFPs. In this study, we used a 3D analyses to investigate the variation in entry points for screw placement inside the OFP of the AC of the pelvis. A universally reproducible entry point could not be determined. Rather, crescent-shaped entry zones were identified, from which screws may be safely placed into the OFP of the AC.

Few studies have investigated screw placement into the AC column. Only two studies investigated the location of entry points. Ochs et al. [2] investigated screw placement into the AC and used a different methodology. They performed spatial measurements in relation to the set landmarks and determined average entry points for males and females. For example, the average entry point in males was located 26.7 ± 3.8 mm (14.0–38.0 mm) lateral of the symphysis and 12.2 ± 1.4 mm (9.0–15.9 mm) below the cranial margin of superior ramus. From a clinical point of view, an average entry point is less useful because of the anatomical variance between cases. Similar to our results, Ochs et al. could not determine a universally reproducible entry point. Unfortunately, the study by Bai et al. was only available in Chinese [6].

Differences between males and females were assessed in both the present study and previous studies. According to Ochs et al., the entry point was located significantly more laterally from the symphysis in females (34.4 ± 5.5 mm versus 26.7 ± 3.8 mm) and closer to the cranial margin of ramus superior ossis pubis (9.9 ± 1.4 mm versus 12.2 ± 1.4 mm). These findings are consistent with the results of our study. The screw diameter was also investigated. In 90% of the males, the anterior column accommodated a 6.5 mm screw. A 6.5 mm screw did not fit one third of the females. These findings are in line with the results of previous studies, which say that the maximum diameter of a screw is statistically significantly larger in males compared to females [2,6,7].

In previous studies on this topic, only one optimal screw was placed per patient. An ideal screw position was defined when the screw was entirely placed within the periacetabular corridor with a central position at the narrowest points of the corridor [8]. This definition of the optimal entry point is arguable. The accessibility of the entry point is not considered when determining the optimal location of the screw. Thereby, it is not proven that a central position in the corridor ensures optimal fixation. A screw may provide more fixation if it is closer to the peripheral bone cortex of the corridor.

In this study, multiple screws were positioned per patient. More than 150 screws were analysed. This illustrated that there are different options for correct screw placement per patient, which allows for greater flexibility during surgery. All possible options for screw insertion are included in the results.

Several limitations require mentioning. Greater accuracy may have been achieved if a larger population was studied. Screws were mostly placed in intact AC. We assume that similar results are achieved in undisplaced fractures. Manually matching 3D models was superior to automated methods but not perfect. Other methods, such as Statistical Shape Modelling (SSM), may arguably result in more accurate matching. However, we expect the difference to be minimal.

Virtual 3D reconstruction models provide practical insight into patient-specific anatomy and can also be useful in preoperative virtual surgery planning [7,9]. Preoperative planned screw position is an accurate alternative for operatively achieved screw position [8]. Patient-specific drilling guides can be used as an adjunct to conventional implants in acetabular fracture surgery, to translate the virtual surgical planning to the operation. The use of patient-specific guides allows for the accurate placement of column screws [10].

## 5. Conclusions

In conclusion, variation exists in the entry point for screw insertion into the anterior column of the pelvis. A universally reproducible entry point does not seem to exist. Interestingly, the location of the entry point is more lateral in females. A patient-specific surgical planning based on a virtual 3D segmentation can be used to provide personalized medicine.

## Figures and Tables

**Figure 1 jpm-12-01748-f001:**
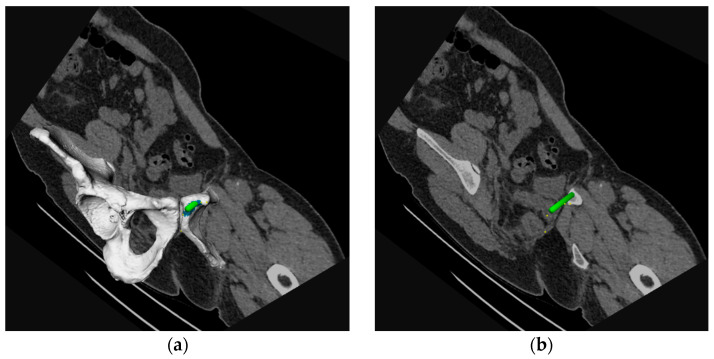
(**a**) A three-dimensional (3D) segmentation of the pelvic bone including a virtually placed column screw (green) with a single slice of the corresponding computer tomography (CT)-scan. (**b**) The position of the screw was assessed by scrolling through the slices of the CT scan.

**Figure 2 jpm-12-01748-f002:**
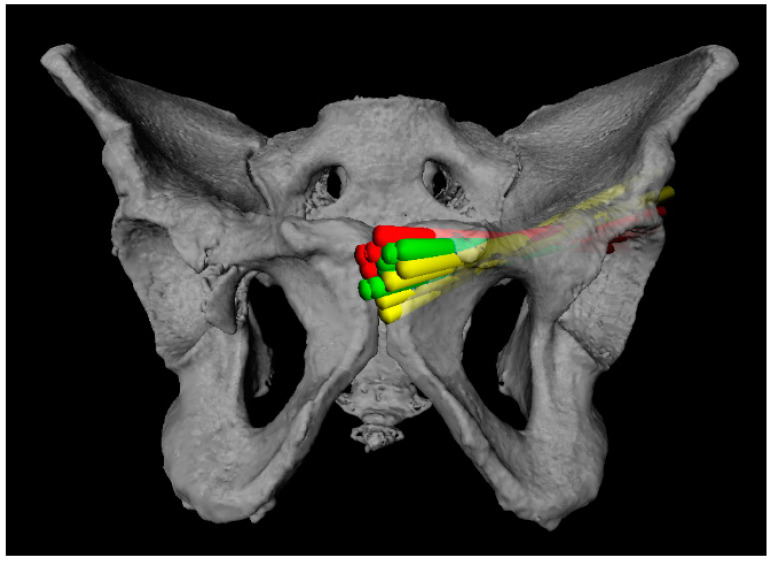
As many screws as possible were placed per case. Green screws were positioned correctly. Yellow screws had an entry point that was deemed surgically dangerous. Red screws exited the OFP prematurely.

**Figure 3 jpm-12-01748-f003:**
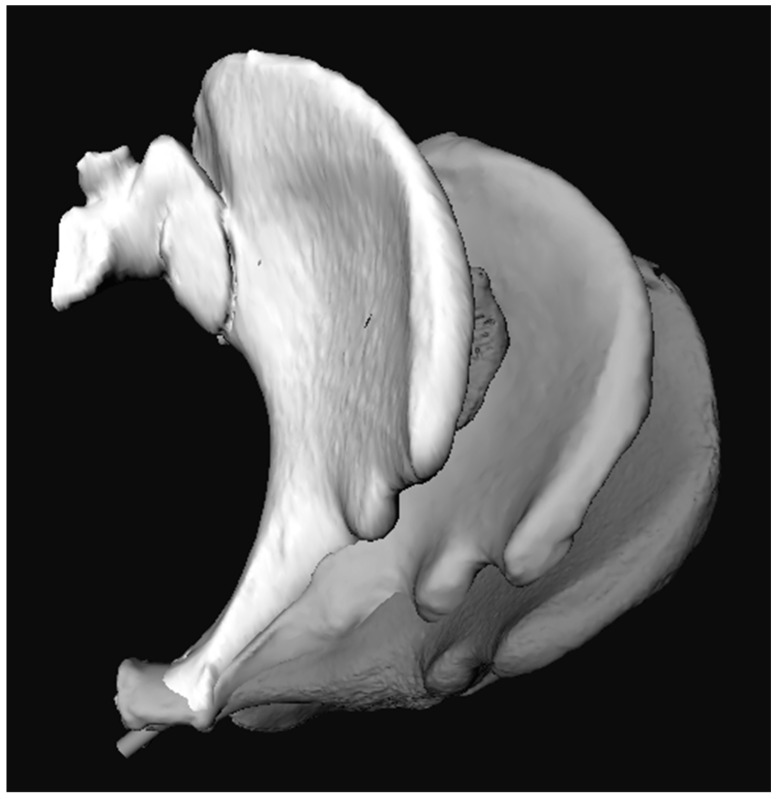
Inlet view of three segmentations of the hemipelvis. Matching was performed using Iterative Closest Point (ICP) or Procrustes analysis, which resulted in inadequate matches. The pubic bones seemed to match adequately, but the osseous fixation pathways (OFPs) diverted.

**Figure 4 jpm-12-01748-f004:**
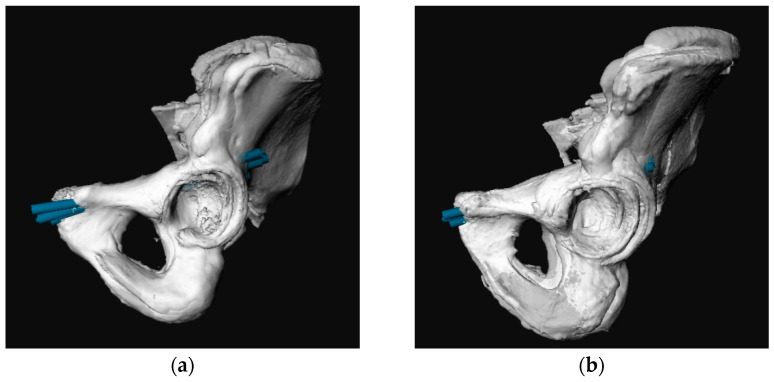
Superimposed three-dimensional (3D) pelvic models including the most optimal screw per case (blue). Analyzing males and females separately resulted in a more compact entry zone. (**a**) sagittal view females; (**b**) sagittal view males; (**c**) frontal view females; (**d**) frontal view males.

**Figure 5 jpm-12-01748-f005:**
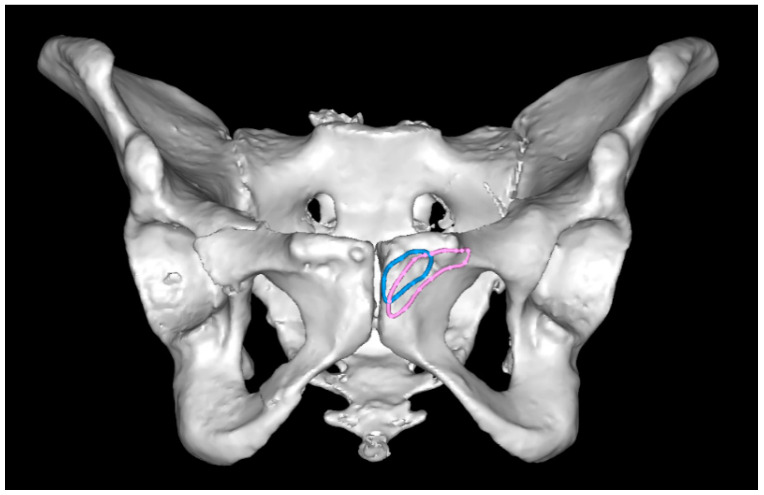
The region of entry points was located more laterally in females (pink) when compared to males (blue).

**Figure 6 jpm-12-01748-f006:**
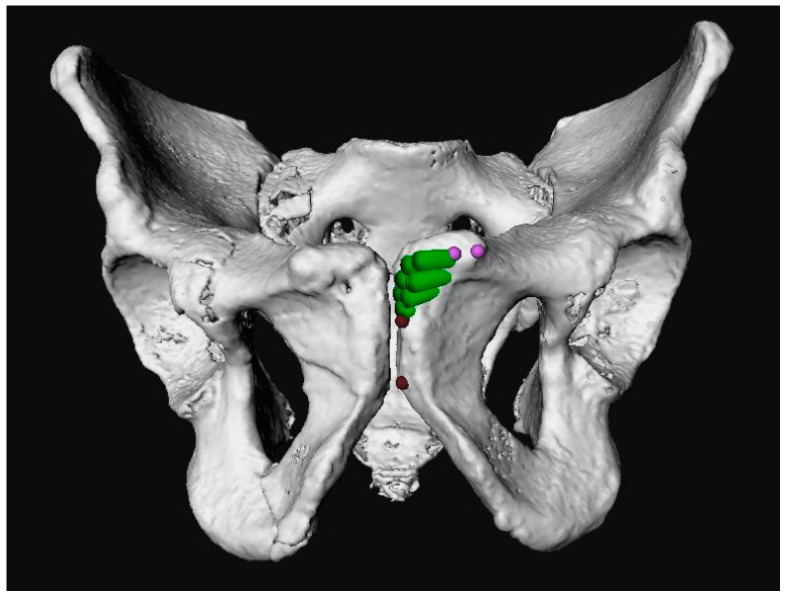
Measurement of the distance between the upper limit of the entry points (green) and the pubic tubercle was measured (pink), and the distance from the lower limit of the entry points (green) to the ramus inferior border (red).

**Table 1 jpm-12-01748-t001:** Patient characteristics.

Variable	Total (*n* = 19)
**Gender**	
Female, *n* (%)	9 (47)
Male, *n* (%)	10 (53)
Median age, years [IQR ^1^]	44 [33]
Median length, cm [IQR]	176 [10]
**Side of fracture**	
Left, *n* (%)	7 (37)
Right, *n* (%)	11 (58)
Bilateral, *n* (%)	1 (5)
**Classification**	
Acetabulum, *n* (%)	4 (21)
Pelvic ring, *n* (%)	15 (79)
Median number of screws per patient, *n* [IQR]	8 [5]

^1^ interquartile range.

**Table 2 jpm-12-01748-t002:** Entry zone analysis.

	Mean (*n* = 19)	Female (*n* = 9)	Male (*n* = 10)
Median upper limit from pubic tubercle, mm [IQR ^1^]	9.8 [5.0]	9.6 [3.2]	11.1 [5.7]
Median lower limit from ramus inferior border, mm [IQR]	17.6 [9.1]	17.6 [8.1]	17.5 [6.9]
Median circumference, mm [IQR]	71.4 [11.8]	74.9 [9.7]	70.1 [16.6]
Median area, mm^2^ [IQR]	312.6 [278.1]	503.8 [280.0]	370.1 [227.0]

^1^ interquartile range.

## Data Availability

The authors declare that the data supporting the findings of this study are available within the paper.

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
