# Peer review of "Entry Point Variation in the Osseous Fixation Pathway of the Anterior Column of the Pelvis—A Three-Dimensional Analysis"

_jpm, 2022, doi:10.3390/jpm12101748_

Round 1

Reviewer 1 Report

The authors have developed a well-conducted and well-written retrospective cohort study to determine the variation in screw insertion entry point location in the anterior pelvic spine fixation pathway.

However, I suggest two clarifications or modifications that will in my opinion improve the quality of their manuscript:

1. Add the research design in the title or delete that it is a cohort study in the "Methods" section.

2. You need, at least, the endorsement or authorization of an Ethics Committee, preferably of your hospital.

3. Between lines 209-214, I recommend mentioning future research directions that also analyze the differences in pre-post surgical blood sampling, as developed in another study, which I recommend citing as a reference:DOI: 10.1097/TGR.0000000000000337

Author Response

Dear editor-in-chief and reviewers,

Thank you for appraisal of our manuscript. We believe that the comments of the reviewer further improve the quality of our report. For that we are grateful.

Below you will find a point-by-point explanation on the details of the revisions to the manuscript and our responses to the referees’ comments.

Reviewer 1

Comment 1

We agree with the suggestion and deleted ‘cohort study’ from the “Abstract” (line 13) and the “Methods” (line 64) section in the manuscript.

Comment 2

In the Netherlands, this type of study (retrospective, anonymized data) does not fall under the scope of the Medical Research Involving Human Subjects Act (WMO), and consequently does not have to be reviewed by an accredited local medical research ethics committee (MREC) or the Central Committee on Research Involving Human Subjects (CCMO). We have added ‘by the local medical research ethics committee’ (lines 57 to 60) in order to improve the clarity of this specific part of the manuscript. We hope the reviewer agrees.

Comment 3

A third comment was provided by the reviewer that refers to an article with the title ‘Impact on Blood Tests of Lower Limb Joint Replacement for the Treatment of Osteoarthritis’ (https://doi.org/10.1097/TGR.0000000000000337). We suspect that this comment has been provided per accident because the subject is not related to the context of our study. Furthermore, the reviewer initially refers to two comments instead of three. We kindly ask confirmation of our suspicion and whether this comment can be ignored. Thank you.

Reviewer 2 Report

Entry point variation of the osseous fixation pathway of the anterior column of the pelvis – A 3-dimensional analysis

the topic chosen by the authors is certainly very modern and of scientific interest.  the study was conducted with meticulousness and a perfect description of the materials and methods.  very interesting results and conclusions of scientific interest.  a correct 3D planning is certainly valid for the correct search of the entry point

Author Response

Dear editor-in-chief and reviewers,

The reviewer did not supply comments and suggestions after their review.

Please contact us in the event of further questions. We will gladly clarify these.

On behalf of the study team we thank you for this opportunity.

With kind regards,

Reviewer 3 Report

It is unusual to be able to read a manuscript with such minute and meticulously written methodology, complete with beautifully combined hypotheses and questions.

            The topic is rarely touched in the literature, with proof being the only 2-3 articles that analysed the same topic.

             Illustrations and tables correctly enhance the results, and the discussion part is well structured. I congratulate the team for providing the literature with these findings.

Author Response

(The authors gave the same response as above.)

Round 2

Reviewer 1 Report

The authors have answered all my questions.

Congratulations for the work.